# Characterizing the Structural and Functional Properties of Soybean Protein Extracted from Full-Fat Soybean Flakes after Low-Temperature Dry Extrusion

**DOI:** 10.3390/molecules23123265

**Published:** 2018-12-10

**Authors:** Wenjun Ma, Fengying Xie, Shuang Zhang, Huan Wang, Miao Hu, Yufan Sun, Mingming Zhong, Jianyu Zhu, Baokun Qi, Yang Li

**Affiliations:** 1College of Food Science & Key Laboratory of Soybean Biology in Chinese Ministry of Education, Northeast Agricultural University, Harbin 150030, China; mwj@163.com (W.M.); spxfy@163.com (F.X.); szhang@neau.edu.cn (S.Z.); whname@neau.edu.cn (H.W.); humiao7890@163.com (M.H.); sunyufan4399@163.com (Y.S.); zmm04261213@126.com (M.Z.) zjy@neau.edu.cn (J.Z.); 2National Research Center of Soybean Engineering and Technology, Harbin 150030, China; 3Harbin Institute of Food Industry, Harbin 150030, China

**Keywords:** full-fat soybean flakes, low-temperature extrusion, soy protein isolates, conformational properties, functional changes

## Abstract

The soy protein isolates (SPI) extracted from different extruded full-fat soybean flakes (FFSF), and their conformational and functional properties were characterized. Overall, the free thiol (SH) content of SPI increased when the extrusion temperature was below 80 °C and decreased at higher temperatures. Soy glycinin (11S) showed higher stability than β-conglycinin (7S) during extrusion. Results also indicated that the increase in some hydrophobic groups was due to the movement of hydrophobic groups from the interior to the surface of the SPI molecules at extrusion temperatures from 60 to 80 °C. However, the aggregation of SPI molecules occurred at extrusion temperatures of 90 and 100 °C, with decreasing levels of hydrophobic groups. The extrusion temperature negatively affected the emulsifying activity index (EAI); on the other side, it positively affected the emulsifying stability index (ESI), compared to unextruded SPI.

## 1. Introduction

Soybean (particularly Glycine max (L.) Merrill) is an exceptional legume because of its extraordinary of oil and protein contents. Soybean oil is mainly extracted by solvent extraction, which not only denatures the protein during the process but additionally affects the quality of the oil too [1,2]. Among the possible existing alternative methods, enzyme-assisted aqueous extraction processing (EAEP) has been contemplated as an alternative preference for immediate oil extraction with no residual solvents and protein, and with lower damage during extraction [3,4].

Potential stages in refining oil extraction through EAEP using oil-bearing materials include pretreatments that can break the cell walls and discharge the oil in the form of either free oil or as an emulsified cream [5]. Therefore, the search for an efficient means of pretreatment that achieves both cell distortion and low denaturation of soybean protein before enzymatic extraction is significant to oil extraction and protein using EAEP.

Pretreatments such as high pressure, ultrasound, and extrusion have been extensively studied [6,7,8]. Extrusion cooking is carried out via a combination of appropriate levels of moisture, temperature, pressure, and mechanical shear [9]. The combination of extrusion and EAEP promotes equally the breaking down of the cell wall and protein denaturation, and seems to support aqueous oil extraction and proteolytic measurements [6,10]. In addition, the benefits of using extrusion treatment depend on the possibility of circumventing the materialization of oil-in-water emulsion, which is currently difficult to disperse after extraction [11]. The flaking and extruding of soy was proved to be a way of increasing oil extraction efficacy through the EAEP of soybeans, resulting in an increase of oil extraction recovery to 88% [7,12]. Increasing oil recovery and protein fraction have been the motivators for EAEP optimization. Nevertheless, when applied to FFSF during extrusion, the temperature, high shear, and the protease carrying during EAEP might be potentially unfavorable to the protein fractions recuperated after the process. The poor solubility and low digestibility of the products were the outcomes of high-temperature extrusion processing, and are limitations of utilizing soy protein in certain foods [13].

Several studies have been carried out regarding the impacts of the extrusion process on the conformational and functional changes of soy at extrusion temperatures beyond 100 °C, which cause severe denaturation of the protein. Under low-temperature extrusion conditions, the impacts of extrusion would be expected to be different from high-temperature extrusion. Consequently, this study aimed to investigate the effect of extrusion temperatures of 60, 70, 80, 90, and 100 °C on the functional and structural properties of soy protein extracted from FFSF after extrusion.

## 2. Results and Discussion

### 2.1. Molecular Weight Distribution Analysis

Figure 1 displays the changes in the MW distributions of SPI samples. The MW of SPI samples changed as the extrusion temperature increased. The peaks before the first dash line (300 kDa) were suspected to be glycinin, β-conglycinin, and 15S, while the peaks below 16 kDa (not explicitly specified in Figure 1) corresponded to 2S, along with subunits of glycinin, β-conglycinin, and other low-MW proteins. The untreated SPI sample had a peak in the MW (range: 100–300 kDa) which matched β-conglycinin. However, this peak disappeared in the extruded SPI samples, indicating that the extrusion-induced denaturation of β-conglycinin occurred at 60 °C, and continued with increased extrusion temperature. This observation was consistent with the results of [14], who reported that extrusion resulted in more advanced denaturation of β-conglycinin than glycinin, causing a reduced β-conglycinin: glycinin ratio. The denaturation of β-conglycinin might have resulted in a yield of higher-MW proteins, or the formation of highly aggregated proteins via non-covalent interaction, such as electrostatic interactions, hydrophobic interactions, and hydrogen bonds. Chen [15] also reported that polymerization similarly occurred during extrusion. The formation of minor peaks between 3 and 10 kDa can be seen in Figure 1, and could be a result of protein hydrolysis. 

### 2.2. SDS-PAGE Analysis

Results obtained from SDS-PAGE of SPI samples are presented in Figure 2. The untreated SPI sample exhibited a typical soy protein profile, in agreement with Mujoo [16], indicating excellent quality of the SPI sample. Protein electrophoresis profile is commonly used as an indicator to elucidate the protein profile, including the presence of abnormal proteins, the absence of standard proteins, and the distribution and concentrations of proteins. According to Figure 2, soy glycinin (11S) showed higher stability than soy β-conglycinin (7S) during extrusion in a temperature range from 60 to 100 °C. Upon increasing the extrusion temperature to 90 and 100 °C, both α and α′ subunits of β-conglycinin were no longer present, as reflected by the absence of colored bands in the resulting gel. This observation suggested that the proteins with 48–100 kDa might be hydrolyzed into proteins of lower MW. While considering the results of MW, this phenomenon indicated that the subunits of β-conglycinin could aggregate during extrusion by covalent or non-covalent bonds. Compared to the untreated SPI sample, a single band appeared at 11–17 kDa in the extruded SPI samples, especially the SPI-Ex90 and SPI-Ex100 samples. The appearance of the new band might be due to the formation of discrete polypeptides during extrusion. In a previous study, the changes in wheat proteins during extrusion were investigated. The fragmentation and aggregation of wheat proteins that took place during extrusion were precisely the same as those revealed by SDS-PAGE [17]. 

### 2.3. Free Sulfhydryl (SH) Content of Proteins

The effect of extrusion temperature on the free SH content of SPI samples is shown in Figure 3. Disulfide (SS) bonds are of importance in stabilizing the protein structure via protein crosslinking. Intra- and/or intermolecular SH-SS interchange reactions are intricate in extrusion treatments. The significantly increased amount of free SH bonds at extrusion temperatures below 80 °C may be ascribed to the breaking of noncovalent bonds between protein molecules [18]. The increase in free SH groups may be credited to (1) the exposure of SH groups, which packed the interior of proteins inside the surface of the proteins; and (2) the disruption of SS bonds, in particular, acidic and basic polypeptides of glycinin within protein molecules [19]. However, by increasing the temperature (>80 °C), the concentration of free SH groups decreased significantly. The decreased free SH content for SPI-Ex90 and SPI-Ex100 suggested that some cysteine could be lost because of SH oxidation, forming sulfur oxidation products other than disulfide bonds [20]. These sulfur oxidation products are barely compressed by 2-mercaptoethanol in the subsequent determination of SH contents. Furthermore, the decreased free SH content can mainly be attributed to aggregation.

### 2.4. FTIR Spectra Analysis

The spectral data of the protein secondary structures of the SPI are given in Figure 4A. The protein secondary structure was reflected in amino I. Changes in the amino I moiety implied the secondary structures of SPI samples, which were changed after extrusion. The α-helix, β-sheet, β-turns and random coil were traced, and their contents are displayed in Table 1. SPI-Ex90 and SPI-Ex100 had much lower contents of α-helix. Notably, the β-sheet content was almost the same. Due to the complex mechanisms of extrusion, as well as the various compositional factors that could directly or indirectly affect protein structures, changes in secondary structure content, cannot be predicted easily. According to the results, it is speculated that the changes in the α-helix content could be due to its natural chemical structure, which was largely stabilized by local interactions, while the β-sheet structure was maintained by long-range contacts, resulting in a greater degree of interference during extrusion [21]. The increase in the β-sheet content has been found to be due to extrusion-induced protein aggregates, resulting in a high amount of β-sheet structures [22]. The aggregation of proteins could be a natural response of the β-sheet in preventing hydrophobic exposure and an increase in entropy. The β-turn content increased with the extrusion temperature, especially from 80 °C to 100 °C, indicating the amount of protein aggregation.

### 2.5. Fluorescence Measurements

The fluorescence spectral data of SPI are presented in Figure 4B. The identification of conformational variations in tertiary structures of proteins can be monitored through the measurement of the fluorescence of tryptophan (Trp), which in turn can be achieved by controlling the fluorescence intensity or maximum emission wavelength (λ_max_) changes of Trp [23]. Trp λ_max_ is very sensitive to its local environment. The shift of λ_max_ of Trp below 330 nm may suggest a hydrophobic environment (mainly the interior of a folded protein), while the change in λ_max_ of Trp above 330 nm indicates that Trp became hydrated, reflecting the loss of structural integrity of the protein molecule. Compared with the untreated samples, SPI-Ex60, SPI-Ex70 and SPI-Ex80 showed blue shifts in fluorescence, indicating the reduced amount of exposed Trp, potentially as a consequence of protein aggregation. However, SPI-Ex90 and SPI-Ex100 showed a red shift, demonstrating that the Trp residues situated in the interior regions of the proteins had been exposed to the hydrophilic solvent environment after extrusion [24]. Notably, the fluorescence intensities for the extruded SPI samples at 60, 70, and 80 °C increased, indicating the movement of hydrophobic groups to the protein surfaces. However, the fluorescence intensities decreased in response to the increased extrusion temperature beyond 80 °C, and this suggests that the exposed hydrophobic groups re-associated or aggregated to form a more stable structure [25]. 

### 2.6. Measurement of Surface Hydrophobicity 

This is an index of the hydrophobic group’s content on the protein molecule surface in interaction with the aqueous polar environment. Interestingly, all the extruded SPI samples showed higher surface hydrophobicity than the untreated SPI sample (Figure 5A). This suggests that proteins were unfolded during the extrusion process, releasing the embedded hydrophobic residues. The surface hydrophobicity of the SPI samples exhibited an increasing trend and subsequent decrease at extrusion temperatures of 90 and 100 °C. A similar observation was reported by Jung and Mahfuz, where extrusion resulted in an increase in the surface hydrophobicity of SPI [2]. The decreased surface hydrophobicity may be attributed to the presence of intermolecular interactions between protein molecules, resulting in a higher number of hydrophobic residues being shifted into regions of greater hydrophobicity [26]. 

### 2.7. Solubility Measurement

The solubility of a protein is the ultimate applied measure of protein, as the situation can indicate the denaturation and aggregation, as well as the effect on functional capabilities. Figure 5B unveils the effects of extrusion on SPI solubility. The untreated SPI sample exhibited the highest solubility. A similar trend in the effects of extrusion temperatures and protein solubility was reported in a previous study [17]. It has been said that the main reason for the decreased solubility is the formation of aggregates of higher molecular weight. As discussed previously in Section 2.3, the decreased solubility of all the extruded SPI samples could be credited to the increase of free SH residues and surface hydrophobicity, resulting in the creation of insoluble protein aggregates. Interestingly, the solubility of SPI-Ex100 was significantly higher, contrary to the trend of decreasing solubility, which could be due to the hydrolysis of proteins. As previously discussed, smaller proteins were detected for SPI-Ex100, as shown in Figure 1 and Figure 2.

### 2.8. Emulsion Properties Analysis

Emulsifying properties of proteins are often assessed by the emulsifying activity index (EAI) and emulsifying stability index (ESI). The EAI is a measure of the ability and capacity of a protein absorbed to the interfacial area of oil and water within an emulsion, i.e., the surface area created per unit mass of the emulsifier. The ESI reflects the time required to achieve turbidity of the emulsion that is one-half of the defined time. The EAI and ESI results are revealed in Figure 5C. The protein’s surface hydrophobicity is regarded as an indicator of the emulsification properties of proteins. However, according to Figure 5C, the EAI results are not correlated with the surface hydrophobicity. Similar findings were found in the famous Food Chemistry book by Fennema, in which no strong correlation was observed between the emulsifying properties and surface hydrophobicity of soy protein [27]. 

The EAI of SPI samples decreased linearly with the increase of extrusion temperatures. As mentioned in Section 2.3, the protein structure unfolded during extrusion due to the heat treatment, resulting in hydrophobic residues or exposed SH. The exposed SH or hydrophobic residues could affect intermolecular bonding between protein molecules via SS and SH interchange reactions or hydrophobic interactions. However, if the interactions between protein molecules are too robust, this could affect the interfacial aggregation, coagulation and eventual precipitation of the protein. These interactions would be detrimental to the emulsification ability of protein [27]. Therefore, the continuous decrease in EAI was attributed to the strong interactions between protein molecules. Furthermore, glycinin has a more compact structure, stabilized by SS bonds, and as a result, its emulsifying ability is weaker than that of β-conglycinin, which has a lower disulfide bond content [28]. The decreased EAI of SPI-Ex90 and SPI-Ex100 could be due to the lack of β-conglycinin. However, the extruded samples showed better ESI than the untreated samples; a more substantial number of exposed SH residues might have helped to stabilize the emulsion, resulting in higher ESI than the untreated samples. The behavior of proteins in forming an emulsion is very complicated and not yet well understood [27]. However, the findings in this section could help to better understand the behavior of the emulsifying properties of SPI under different extrusion processes. 

### 2.9. Statistical Correlation Analysis

The correlations between the secondary structure changes and functionalities of SPI were analyzed using a robust open-source software platform (R-project, http://www.r-project.org). As shown in Figure 6, the histograms representing the distribution of each variable are shown along the diagonal line. The bivariate scatter plots with a fitted line are presented in the lower right triangle, and the values of correlation, with significance levels shown as stars, are shown in the upper left triangle. The level of −0.81 indicates that the α-helix content was significantly negatively correlated (*p* < 0.05) with β-sheet content. The α-helix, β-sheet and β-turns were found to dramatically affect the solubility (S) and EAI of SPI. 

## 3. Materials and Methods

### 3.1. Materials

Full-fat soybean flakes (cultivar Dong-Nong 42 harvested in Harbin, China in 2017). This cultivar contained 20% fat and 40% crude protein on a dry basis. The flakes were airtight and kept in plastic flasks at 4 °C until further usage. Both 1-anilino-8-naphthalenesulphonate (ANS) and 5,5′-Dithiobis-(2-nitrobenzoic acid) (DTNB) were obtained from Sigma-Aldrich (St. Louis, MO, USA). 

### 3.2. Extrusion Pretreatment

A co-rotating twin-screw extruder (Evolum 25, Clextral, Firminy, France) using a feeder (model T20, K-TRON (Schweiz) AG, Hillenbrand, France) was used in all extrusion processes. The barrel contained six individual temperature control sections. The dimension to diameter ratio of the extruder barrel was 32:1, while the screw had eleven conveying and kneading screws. Both screw and feed rates remained constant at 300 rpm and 50 g/min, respectively. To initiate the extrusion, soybean flakes were first ground into a powder using a laboratory grinder (FW-100, Shaoxing Kehong Instrument Co., Ltd., Zhejiang, China) and subjected to a 60-mesh strainer. The barrel temperatures were set at a fixed temperature gradient of 30, 30, 40, 40, and 50 °C. Then, the flake powder was processed at die temperatures of 60, 70, 80, 90, and 100 °C, respectively. The length of each temperature zone was 10 cm, and the die diameter was 1 cm. The extrusion speed was automatically controlled, such that no extrusion took place until the desired extrusion temperature had been reached. The extruder’s moisture content was attuned to 16% by directly adding water in the first soybean powder. The extrudate was collected under identical settings, which were confirmed by the constant die temperatures. The products were cooled to room temperature before being pulverized into a fine powder (<250 μm). The extruded soybean powder (ESP) was taped up and kept at 4 °C in plastic bags until subsequent analysis.

### 3.3. Soy Protein Isolate Preparation

The ESPs from different extrusion treatments were subjected to the extraction of soy protein isolate (SPI) [29]. Briefly, the ESP was defatted using hexane. Water (3 L) was blended with the defatted ESP (200 g) to form a mixture, followed by adjusting its pH to 8.0 using NaOH (2.0 M). The blend was mixed at 25 °C with continuous stirring for 2 h, and was centrifuged (Hunan Changsha Xiangyi Centrifuge Instrument Co. Ltd., Changsha, China) at 10,000× *g* for 20 min. The aqueous supernatant was taken cautiously, and the pH was attuned to 4.5 using HCl (2.0 M) to precipitate the protein. Centrifugation (10 min at 10,000× *g*) was conducted again to separate the acid-precipitated protein. The acid-precipitated protein was washed four times using water and neutralized using NaOH (2.0 M) to form an aqueous phase containing solubilized proteins. The aqueous phase was removed in a freeze dryer (FEtseries, GOLD SIM, Newark, NJ, USA) to obtain the SPI. The extraction rate of SPI was approximately 70%. The overall protein content of the SPI was above 90%, as measured by the Kjeldahl method. 

### 3.4. Molecular Weight (MW) Distribution Analysis

The MW distributions of the SPI samples were analyzed using high-performance size exclusion chromatography [30]. An ÄKTA purifier system equipped with a HiLoadTM 16/60 SuperdexTM 200 prep grade column (GE Healthcare Bio-Sciences AB, Uppsala, Sweden) was used to perform the analysis. SPI or extruded samples were dissolved in water to a concentration of 1 mg/mL, and were centrifuged at 10,000× *g* at 25 °C for 10 min. The resulting mixture was strained via a cellulose acetate membrane (0.22 μm). Injection volumes were set at 2 mL, and the flow rates were set at 1 mL/min. The mobile phase was phosphate-buffered saline (50 mM PO_4_^3−^ and 300 mM NaCl) at pH 7.4. The flow rate was examined at 280 nm. MW distributions were determined using an external calibration curve covering MW from 65 to 200 kDa. 

### 3.5. Sodium Dodecyl Sulfate-Polyacrylamide Gel Electrophoresis (SDS-PAGE) Analysis

The SDS-PAGE analysis was performed using a discontinuous buffer method (reducing agent: 2-mercaptoethanol) [31]. The running gel was prepared by pouring 5% stacking gel on top of 15% separating gel. SPI samples were added to the SDS sample buffer (0.0625 mol/L Tris-HCl, 10% SDS, 10% glycerin, 5% 2-mercaptoethanol and 0.0025% bromophenol blue) at a concentration of 2 mg/mL. The mixtures were incubated at room temperature for 1 h, heated to 95 °C in a water bath for 5 min, then cooled to atmospheric temperature. Electrophoresis was performed using 15 μL aliquots of the mixtures, and the resulting gels were stained with Coomassie Brilliant Blue R-250 (Gel Doc™, Bio-Rad, California, USA) (methanol:water:acetic acid, 5:4:1) (*v*/*v*) for 30 min.

### 3.6. Free Sulfhydryl (SH) Content Analysis

SPI samples were dissolved in an EDTA-Tris-glycine buffer (0.09 M glycine, 0.086 M Tris, and 4 mM Na_2_EDTA with pH 8.0) at a concentration of 0.2% (*v*/*v*). The samples were incubated at 25 °C for 24 h in a water bath with continuous shaking and were finally centrifuged at 10,000× *g* for 15 min at 4 °C. The supernatant was collected for SH group content analysis. Ellman’s reagent solution (0.03 mL) was added to a 3 mL aliquot of the sample and was mixed quickly before being allowed to stand for 15 min at 20 °C. Absorbance was measured at 412 nm [32]. An SP-721 UV spectrophotometer (Lengguang Technology Co., Ltd., Shanghai, China) was auto-zeroed using a blank sample (Tris-glycine-EDTA buffer). Calculations were based on an extinction coefficient of 13,600 M^−1^ cm^−1^ for the thiolate chromogen using the following equations:A_412_ = A _(with DTNB)_ − A_(without DTNB)_(1)
μM SH (g) = 73.53A_412_*D*/*C*(2)
where A_412_ is the absorbance at 412 nm; *C*. is the sample concentration in mg solids/mL; *D* is the dilution factor 2.03. The constant of 73.53 was derived from 106/(1.36 × 104), for which 1.36 × 104 was the molar absorptivity and 106 was for unit conversions (molar basis to μM/mL, and from mg solids to g solids).

### 3.7. Measurement of Fourier Transform Infrared (FTIR) Spectra

FTIR spectroscopy was implemented accordingly [33]. D_2_O was used to prepare SPI sample solutions, due to the greater transparency of D_2_O in the infrared region compared to H_2_O. SPI sample solutions were prepared by dissolving SPI samples in a pH 7.0 phosphate buffer (0.01 M) at a concentration of 0.1 g/mL. To confirm complete D-H exchange, SPI sample solutions were prepared one day in advance. SPI sample solutions were placed in an IR cell with 25 mm path length CaF_2_ windows. Sixty-four scans were performed at a resolution of 4 cm^−1^. Infrared spectra of SPI sample solutions were recorded using a MAGNA-IR560 spectrometer (Nicolet Co., Ltd., Madison, WI, USA) at 400 to 4000 cm^−1^. Infrared deconvolutions were executed with the Peakfit software, ver. 4.12 (Sea Solve Software Inc., Los Angeles, California, USA). The half-bandwidth employed for deconvolution was 5 cm^−1^. Band preps within the amide I region (1600–1700 cm^−1^) were performed as per the method in [34,35]. Secondary structure composition was analyzed by Gaussian curve-fitting of an FTIR spectrum (Shimadzu Co., Ltd., Kyoto, Japan) using the Peakfit software. 

### 3.8. Fluorescence Spectra Measurements

Fluorescence measurement was performed based on the method reported by [36] using a Fluor photometer (Hitachi Co., Tokyo, Japan). SPI sample solutions were prepared in a pH 7.6 phosphate buffer (0.01 M) at a concentration of 0.2 mg/mL. The excitation wavelength applied was 295 nm. In addition, the emission bands were logged starting at 300–400 nm, implementing a 5 nm slit for all the emissions and excitations. 

### 3.9. Surface Hydrophobicity Measurement

The SPI samples were dissolved in a pH 7.6 phosphate buffer at 1 mg/mL and centrifuged at 8000× *g* for 20 min at 4 °C [37]. The supernatants were composed, and the soybean protein concentration was measured by the Lowry method. The supernatants were serially diluted in 0.01 mol/L phosphate buffer (pH 7.6) to make various concentrations. Fifty microliters of ANS (8.0 mmol/L in 0.01 mol/L phosphate buffer, pH 7.6) was added to 10 mL of diluted samples, respectively. The fluorescence intensity (FI) of the sample solutions was monitored at 330 and 490 nm for excitation and emission (respectively) through an F4500 fluorescence spectrophotometer (Hitachi Co., Japan). The preliminary slope of protein concentration versus FI plot was used as an index of H_0_.

### 3.10. Solubility Measurement

The protein sample (100 mg) was blended with 10 mL of phosphate buffer solution (0.01 mol/L, pH 7.0), was stirred for 30 min, and centrifuged at 12,000× *g* at 20 °C for 20 min. Crude protein contents within the supernatants were measured by the Lowry technique [32]. Bovine serum albumin ran as a standard. Solubility was expressed as percentages of the original protein within the supernatant.

### 3.11. Emulsion Properties

At a concentration of 1 mg/mL, samples were diluted in 0.1 M phosphate buffer (pH 6.5). Oil-in-water (O/W) emulsions of soybean protein were concocted by adding soluble protein solution (8 mL) to soybean oil (2 mL) before homogenizing in a Fluko homogenizer (Shanghai, China) for 1 min at 20,000 rpm [38]. Both emulsifying activity index (EAI) plus emulsion stability index (ESI) were determined by Equations (3) and (4):(3)$$\mathrm{EAI}\;\left(m^{2}/g\right)=\frac{2\times 2.303\times A_{0}\times DF}{c\times \varphi \times \left(1-\theta \right)\times 10,000}$$
(4)$$\mathrm{ESI}\;\left(\min\right)=\frac{A_{0}}{A_{0}-A_{10}}\times 10$$
where A_0_ and A_10_ are the emulsion absorbance at 0 and 10 min; *θ* is the oil fraction used to form the emulsion (0.25); *DF* is the dilution factor; *φ* is the optical path (0.01 m), and c is the protein concentration in the sample solution (g/mL).

### 3.12. Statistical Analysis

All the experiments were conducted in triplicate. Results were presented as mean ± standard deviation. All analyses were subjected to one-way analysis of variance (ANOVA) at *p* < 0.05 using SPSS software (version 17.0, SPSS Inc., Chicago, IL, USA).

## 4. Conclusions

The results of this research suggested that extrusion treatment induced molecular aggregation and dissociation of SPI subunits, exposed hydrophobic groups, and buried SH inside. The continuous increases in extrusion temperature resulted in negative impacts on emulsifying activity and solubility. These results may be of great importance in understanding the conformational and functional changes of SPI during extrusion, thereby assisting in the development of soy-based food products with desirable functionalities. The impact of hybrid extrusion and EAEP on the conformational and functional changes of soybean protein remains unknown, but further study is underway in the authors’ lab.

## Figures and Tables

**Figure 1 molecules-23-03265-f001:**
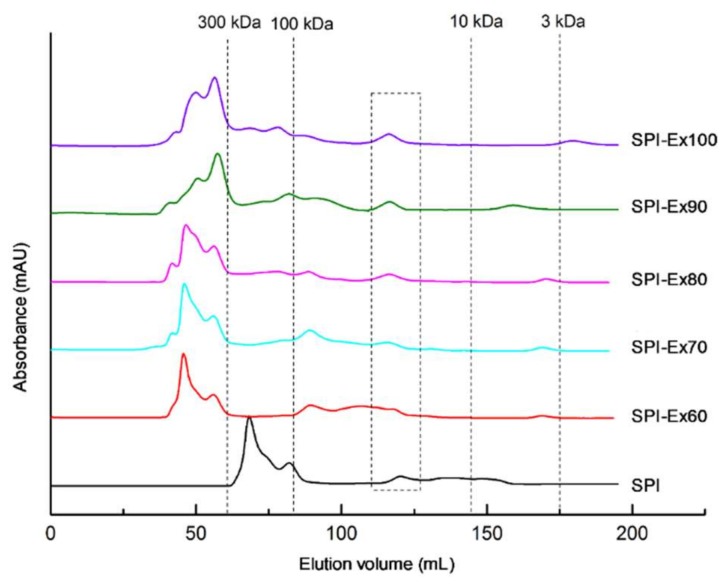
Molecular weight distributions of SPI extracted from full-fat soybean flakes powder with and without extrusion treatment.

**Figure 2 molecules-23-03265-f002:**
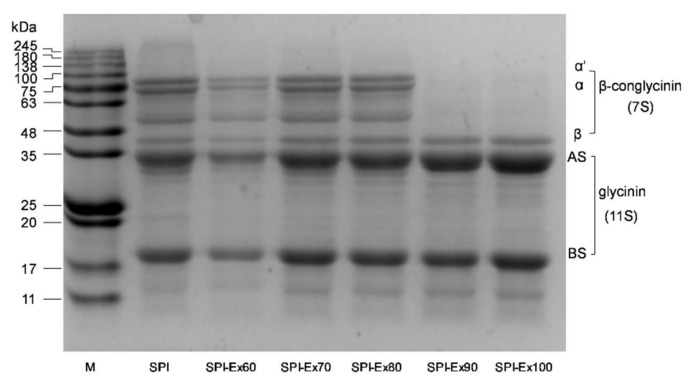
SDS-PAGE images of SPI extracted from full-fat soybean flakes powder with and without extrusion treatment.

**Figure 3 molecules-23-03265-f003:**
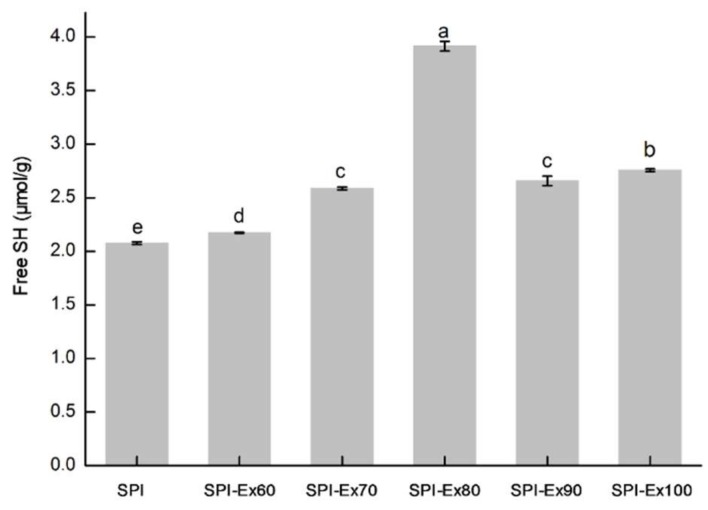
SDS-PAGE images of SPI extracted from full-fat soybean flakes powder with and without extrusion treatment.

**Figure 4 molecules-23-03265-f004:**
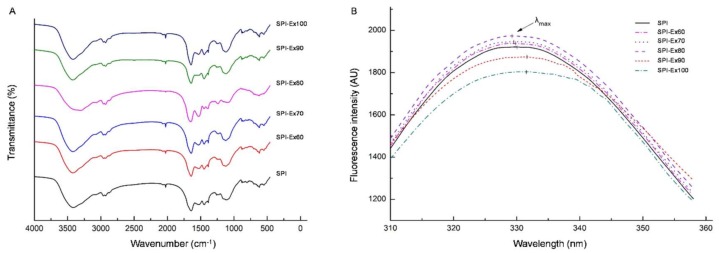
FTIR spectra (**A**) and fluorescence spectra (**B**) of SPI extracted from full-fat soybean flakes powder with and without extrusion treatment.

**Figure 5 molecules-23-03265-f005:**
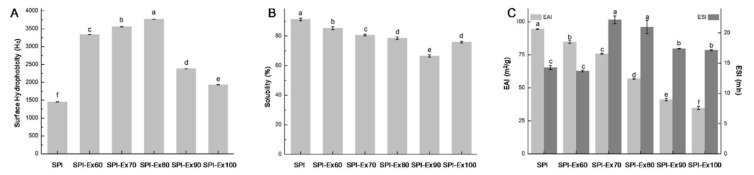
(**A**) surface hydrophobicity, (**B**) solubility, and (**C**) EAI and ESI of SPI extracted from full-fat soybean flakes powder with and without extrusion treatment.

**Figure 6 molecules-23-03265-f006:**
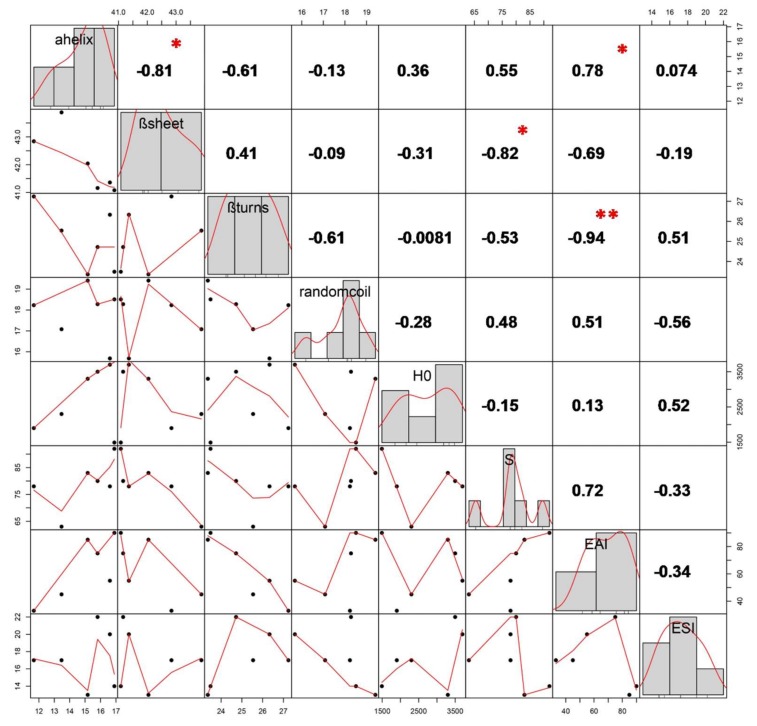
Correlation between the secondary structures composition and functional properties SPI extracted from full-fat soybean flakes powder with and without extrusion treatment. ** and * represent different significance levels at 0, 0.001, and 0.01, respectively.

**Table 1 molecules-23-03265-t001:** Effects of extrusion treatments on the secondary structure of SPI.

	Secondary Structure Composition (%)			
	α-Helix	β-Sheet	β-Turns	Random Coil
untreated	16.90 ± 0.04 ^a^	41.08 ± 0.07 ^a^	23.49 ± 0.04 ^e^	18.51 ± 0.05 ^a,b^
SPI-Ex60	15.17 ± 0.04 ^b^	42.04 ± 0.06 ^a^	23.36 ± 0.05 ^e^	19.41 ± 0.07 ^a^
SPI-Ex70	15.81 ± 0.02 ^b^	41.16 ± 0.03 ^a^	24.72 ± 0.03 ^d^	18.28 ± 0.07 ^a^
SPI-Ex80	16.61 ± 0.08 ^a^	41.36 ± 0.09 ^a^	26.33 ± 0.03 ^b^	15.68 ± 0.08 ^c^
SPI-Ex90	13.47 ± 0.01 ^c^	43.88 ± 0.08 ^b^	25.54 ± 0.09 ^c^	17.08 ± 0.09 ^b^
SPI-Ex100	11.67 ± 0.02 ^d^	42.84 ± 0.00 ^c^	27.24 ± 0.07 ^a^	18.23 ± 0.10 ^a,b^

^a–e^ Mean values in the same column followed by different superscripts in the same column are significantly different (*p* < 0.05).

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
