# Peer review of "Characterizing the Structural and Functional Properties of Soybean Protein Extracted from Full-Fat Soybean Flakes after Low-Temperature Dry Extrusion"

_molecules, 2018, doi:10.3390/molecules23123265_

Round 1
Reviewer 1 Report
This manuscript by Wenjun Ma et al. titled “Characterizing the Structural and Functional Properties of Soybean Protein Extracted from Full-Fat Soybean Flakes after Low-Temperature Dry Extrusion” describes to find out the correlation between the extrusion temperature and the stability of SPI. However, there are some issues in the manuscript which need to be rectified before the acceptance of the paper. Some suggestions are provided below to improve the manuscript quality.
1. In the molecular weight distribution analysis, the peak profile, area, and the relative position are similar between SPI and SPI-Ex60. Their difference is only the displacement in the elution volume. Why?
2. In FTIR spectra analysis, the author mentioned that “the β-sheet content was almost the same” in line 120. However, the author also made a conclusion that the extrusion induced protein aggregates and resulted in a high amount of β-sheet structure. It seems to be inconsistent.
3. In table 1, the content of β-turn was increased with the extrusion temperature especially from 80℃ to 100℃. Whether it is related to the amount of protein aggregation?
Author Response
Point 1: In the molecular weight distribution analysis, the peak profile, area, and the relative position are similar between SPI and SPI-Ex60. Their difference is only the displacement in the elution volume. Why?
Response 1: It proved that extrusion denatured β-conglycinin and resulted in a yield of higher MW proteins.
Point 2: In FTIR spectra analysis, the author mentioned that “the β-sheet content was almost the same” in line 120. However, the author also made a conclusion that the extrusion induced protein aggregates and resulted in a high amount of β-sheet structure. It seems to be inconsistent.
Response 2: Sorry for the not correctly expressed, the β-sheet content was almost the same range from 60 to 80℃.
Point 3: In table 1, the content of β-turn was increased with the extrusion temperature especially from 80℃ to 100℃. Whether it is related to the amount of protein aggregation?
Response 3: Yes, we added the results in the paper.
Reviewer 2 Report
Characterizing the structural and functional properties of Soybean protein extracted from full-fat soybean flakes after low-temperature dry extrusion
The authors in this report (Ma et al) used an extrusion method associated with a low temperature range to extract soy protein isolates. They observed aggregation of SPI at particular higher temperatures (like 90-100 oC).
However, I have few questions and concerns regarding this study.
1. in the SDS-PAGE you see a reduction in the total proteins with SPIEx60. Is it a sampling error? does it mean something?
2. why do you think that there is a complete loss in the 7s protein bands at 90 and 100 temperatures (in Figure 2) ?. In the Chen et al paper (a paper you cited) even at very high temperatures, they were able to see the 7S protein. They saw a reduction in total proteins with low moisture content. Do you think if you have chosen a higher moisture content you could have overcome this situation?
3. If SPIEX80 gives you a higher free SH, do you think that's the best temperature to work?
4. given that the SPI that generated without extrusion treatment has low hydrophobicity, high solubility, and less aggregation what would be your overall conclusion. don't you think you need to change some parameters like moisture content to get the ideal temperature to work with?
Author Response
Point 1: in the SDS-PAGE you see a reduction in the total proteins with SPIEx60. Is it a sampling error? does it mean something?
Response 1: The sample we extracted with different protein contents, and the protein content of SPIEx60 is lower than others. The SDS-PAGE is a qualitative analysis for studying the stability of different subunits.
Point 2: why do you think that there is a complete loss in the 7s protein bands at 90 and 100 temperatures (in Figure 2) ?. In the Chen et al paper (a paper you cited) even at very high temperatures, they were able to see the 7S protein. They saw a reduction in total proteins with low moisture content. Do you think if you have chosen a higher moisture content you could have overcome this situation?
Response 2: In the Chen et al paper, they chose a wide moisture rage, the water in soy powder can protect the protein in case of denaturation. While in our study we attuned the moisture to 16%,which is a relatively low moisture. With a higher temperature(90 and 100) the 7S protein bands disappear.
Point 3: If SPIEX80 gives you a higher free SH, do you think that's the best temperature to work?
Response 3: Yes, with higher free SH the conformation of protein is more unfolded. It’s a good property for food processing.
Point 4: given that the SPI that generated without extrusion treatment has low hydrophobicity, high solubility, and less aggregation what would be your overall conclusion. don't you think you need to change some parameters like moisture content to get the ideal temperature to work with?
Response 4: Thanks for your suggestion, we will change the moisture in our next paper.
Reviewer 3 Report
The article: “Characterizing the structural and functional properties of soybean protein extracted from full-fat soybean flakes after low-temperature dry extrusion” study the impact of extrusion temperature (as pretreatment in oil extraction) over the structure and functionality of soybean protein.
My general opinion is that the work is well described and properly organized. It has a very useful approach because of the impact of pre-treatment of soybean flakes, first in the oil extraction, but also in the properties of proteins used as isolate in the food industry.
My only recommendation for future works is that authors explore the effect of low temperature extrusion also in the oil extraction and not only in protein structure and functionality, despite this, the information included in this article is interesting and useful.
Some specific comments following:
· Title: in the title, the word, characterizing does not describe the specific analysis the authors made. My suggestions is to include some keywords as functionality, or emulsifying activity, and also of course, secondary and tertiary structure in the protein.
· Results and discussion:
o Line 101, “treatmentS”
o Figure 3’s caption is misplaced.
o Line 139, “)”
· Materials and methods:
o Some missed subscripts or superscripts as: phosphate in line 260, line 274 and line 280
o Statistical analysis: the correlation analysis, described in results and discussion (section 2.9) is recommended to be included in here, and improve section 2.9 to discuss more about the interesting correlation found by authors (figure 6).
Author Response
Point 1: Title: in the title, the word, characterizing does not describe the specific analysis the authors made. My suggestions is to include some keywords as functionality, or emulsifying activity, and also of course, secondary and tertiary structure in the protein.
Response 1: We did corrections in our paper.
Point 2: Results and discussion:
o Line 101, “treatmentS”
o Figure 3’s caption is misplaced.
o Line 139, “)”
Response 2: We did corrections in our paper.
Point 3: Materials and methods:
o Some missed subscripts or superscripts as: phosphate in line 260, line 274 and line 280
o Statistical analysis: the correlation analysis, described in results and discussion (section 2.9) is recommended to be included in here, and improve section 2.9 to discuss more about the interesting correlation found by authors (figure 6).
Response 3: We did corrections in our paper.
